# Lysosomal cholesterol export reconstituted from fragments of Niemann-Pick C1

Michael Nguyen Trinh[1†], Michael S Brown[1*], Joachim Seemann[2], Joseph L Goldstein[1*], Feiran Lu[1†]

[1]Departments of Molecular Genetics, University of Texas Southwestern Medical Center, Dallas, United States; [2]Cell Biology, University of Texas Southwestern Medical Center, Dallas, United States

**Abstract** Niemann-Pick C1 (NPC1) is a polytopic membrane protein with 13 transmembrane helices that exports LDL-derived cholesterol from lysosomes by carrying it through the 80 Å glycocalyx and the 40 Å lipid bilayer. Transport begins when cholesterol binds to the N-terminal domain (NTD) of NPC1, which projects to the surface of the glycocalyx. Here, we reconstitute cholesterol transport by expressing the NTD as a fragment separate from the remaining portion of NPC1. When co-expressed, the two NPC1 fragments reconstitute cholesterol transport, indicating that the NTD has the flexibility to interact with the remaining parts of NPC1 even when not covalently linked. We also show that cholesterol can be transferred from the NTD of one full-length NPC1 to another NPC1 molecule that lacks the NTD. These data support the hypothesis that cholesterol is transported through interactions between two or more NPC1 molecules.
DOI: https://doi.org/10.7554/eLife.38564.001

*For correspondence:
mike.brown@utsouthwestern.edu
(MSB);
joe.goldstein@utsouthwestern.
edu (JLG)

†These authors contributed
equally to this work

Competing interests: The
authors declare that no
competing interests exist.

Reviewing editor: William A
Prinz, National Institutes of
Health, United States

## Introduction

Cholesterol enters mammalian cells through the receptor-mediated uptake of plasma low density lipoprotein (LDL) (*Brown and Goldstein, 1986*). After its liberation from LDL in lysosomes, cholesterol is bound by a soluble intralysosomal carrier protein designated Niemann-Pick C2 (NPC2), which delivers it to a membrane-embedded transport protein designated NPC1 (*Pentchev, 2004*). The mechanism by which NPC1 transfers cholesterol across the lysosomal membrane is beginning to be elucidated through a combination of studies that are functional (*Infante et al., 2008*; *Wang et al., 2010*; *Deffieu and Pfeffer, 2011*) or structural (*Kwon et al., 2009*; *Gong et al., 2016*; *Li et al., 2016a*, *2016b*, *2017*). Successful elucidation of this transport mechanism will provide a paradigm for the transport of a lipid across a membrane, and it will also advance the understanding of Niemann-Pick C disease, a fatal lysosomal lipid storage disease caused by mutations in NPC1.

NPC1 is a complex protein of 1278 amino acids with 13 transmembrane helices and three structured lumenal domains (*Figure 1*). Cholesterol-loaded NPC2 binds to one of these domains designated the middle lumenal domain (MLD), which positions NPC2 so that it can transfer its cholesterol to the N-terminal domain (NTD) (*Deffieu and Pfeffer, 2011*). The NTD then must transfer its cholesterol across the ~80 Å glycocalyx so that it can reach the membrane domain of NPC1 (*Neiss, 1984*). This transfer is believed to require the interaction of the NTD with a 9-amino acid loop designated the Ω loop in the C-terminal domain (CTD) (*Li et al., 2017*). The membrane domain of NPC1 contains a sequence of five transmembrane helices that is shared with other membrane proteins whose actions are related to cholesterol (*Brown et al., 2018*). This sequence, termed the sterol-sensing domain (SSD), contains an intramembrane cleft that has been proposed to bind cholesterol and may function in the transport reaction (*Li et al., 2016b*).

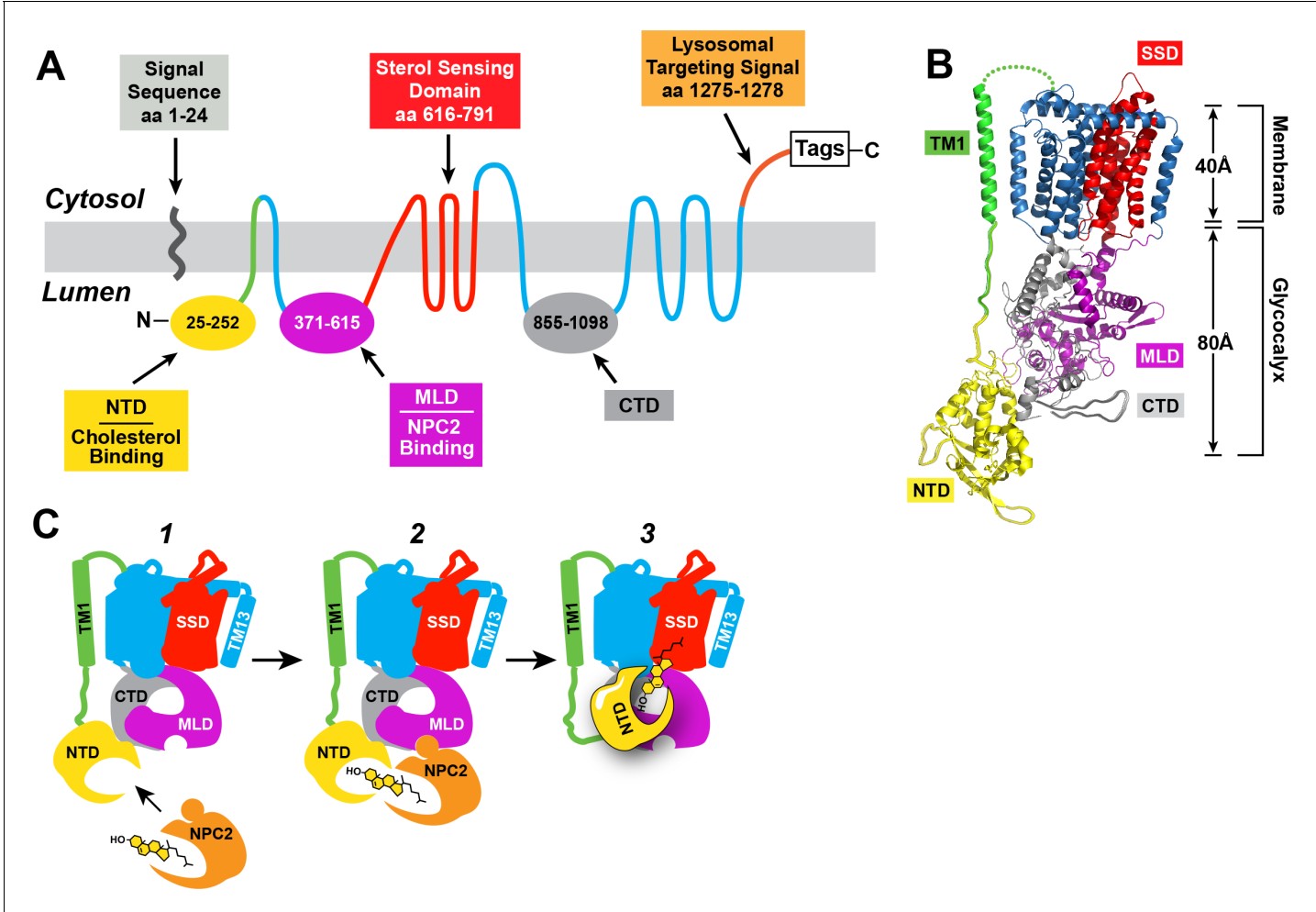

**Figure 1.** Human NPC1: topology, structure, and function. (**A**) Predicted topology of human NPC1 based on the data of *Davies and Ioannou (2000)*. Functional domains of the protein are shown in different colors. NTD, N-terminal domain; MLD, middle lumenal domain; CTD, C-terminal domain. (**B**) Structure of full-length NPC1 as determined by cryo-electron microscopy (*Gong et al., 2016*) was accessed from the Protein Data Bank (PDB: 3JD8) and color-matched to the topology map in (**A**). Image was generated using the PyMOL Molecular Graphics System, Version 2.0 Schrödinger, LLC. TM1, transmembrane helix 1; SSD, sterol-sensing domain. (**C**) Model for cholesterol transfer. (1) NPC2 brings cholesterol to NPC1, which is embedded in the lysosomal membrane. (2) NPC2 binds to the MLD of NPC1 and transfers its cholesterol to the NTD of NPC1 in a hydrophobic handoff (*Kwon et al., 2009*; *Deffieu and Pfeffer, 2011*). (3) Cholesterol is transferred from the NTD to the membrane – embedded SSD.
DOI: https://doi.org/10.7554/eLife.38564.002

The current studies were designed to further our understanding of the mechanism by which the NTD transfers its cholesterol to the membrane domain of NPC1. Our data demonstrate that the NTD can accomplish this transfer even when it is produced as a separate protein that is not linked covalently to the remainder of NPC1. Moreover, we show that the NTD of one full-length NPC1 protein can transfer its cholesterol to a separate NPC1 protein that does not contain an NTD, indicating that cholesterol transfer may normally occur through the interaction of two or more NPC1 molecules. Considered together with previous data, the current studies begin to provide a glimpse into the mechanism by which a lipid can be transferred across a glycocalyx and through a membrane.

## Results

*Figure 1A* shows the membrane topology of NPC1 with its three large lumenal loops designated N-terminal domain (NTD), middle lumenal domain (MLD), and C-terminal domain (CTD). The MLD and CTD flank the sterol-sensing domain (SSD), which is composed of five membrane spanning

helices separated by short connecting loops. *Figure 1B* shows the structure of NPC1 as determined by cryo-electron microscopy to 4.4 Å resolution (*Gong et al., 2016*). The NTD is connected to the membrane by a long stalk that extends far enough to cross the ~80 Å glycocalyx that shields the lysosomal membrane from lysosomal lipases and proteases. *Figure 1C* illustrates a proposed mechanism by which NPC1 facilitates cholesterol transport out of the lysosome. Cholesterol released from LDL is bound to NPC2 (*step 1*), which binds to the MLD of NPC1, which orients the NPC2 so that it can transfer its cholesterol to the NTD by a sliding mechanism (*step 2*) termed the hydrophobic handoff (*Kwon et al., 2009*). The next step is the transfer of cholesterol from the lumenal NTD through the glycocalyx to reach the membranous SSD (*step 3*). This transfer would require a large conformational change in the protein to re-orient the NTD so that it approaches the membrane. A recent structural analysis of NPC1 by X-ray crystallography at 3.3 Å resolution demonstrated that the NTD binds to the CTD (*Li et al., 2017*). Binding is mediated by a short loop designated the Ω loop that projects from the CTD and binds to a region of the NTD. When the Ω loop was deleted from the CTD, the mutant protein traveled normally to lysosomes, but it failed to transport cholesterol, suggesting that this interaction is crucial for the transfer of cholesterol from the NTD to the membrane (*Li et al., 2017*).

To further define the mechanism for cholesterol transfer from the NTD, we sought to determine whether the NTD could still interact with the remainder of NPC1 if the NTD was produced as a separate protein. For this purpose, we created a plasmid encoding the NTD linked to the membrane by a single transmembrane helix that corresponds to the first transmembrane helix (TM1) of NPC1, which normally follows the NTD (*Figure 2A*). In this plasmid TM1 is followed by a lysosomal targeting sequence (*Watari et al., 1999*) and two epitope tags, Flag and StrepTactin. We also created a plasmid designated pΔNTD that encodes the remainder of NPC1 that is missing the NTD (*Figure 2B*).

To determine whether the NPC1 fragments would reach the lysosome, we used immunofluorescence microscopy (*Figure 3*). We introduced plasmids encoding Flag-tagged versions of NPC1 into SV589 cells, a line of SV40-immortalized human fibroblasts (*Yamamoto et al., 1984*). We fixed and permeabilized the cells and visualized NPC1 by incubation with an antibody against the Flag epitope tag followed by a second antibody conjugated to a dye that fluoresces green. Lysosomes were visualized with anti-LAMP-2 directed against a lysosomal protein followed by a second antibody that fluoresces red. Full length NPC1, ΔNTD, and NTD-TM1 all localized to lysosomes as indicated by the yellow color in the merged images. In *Figure 3* we also studied the localization of three other

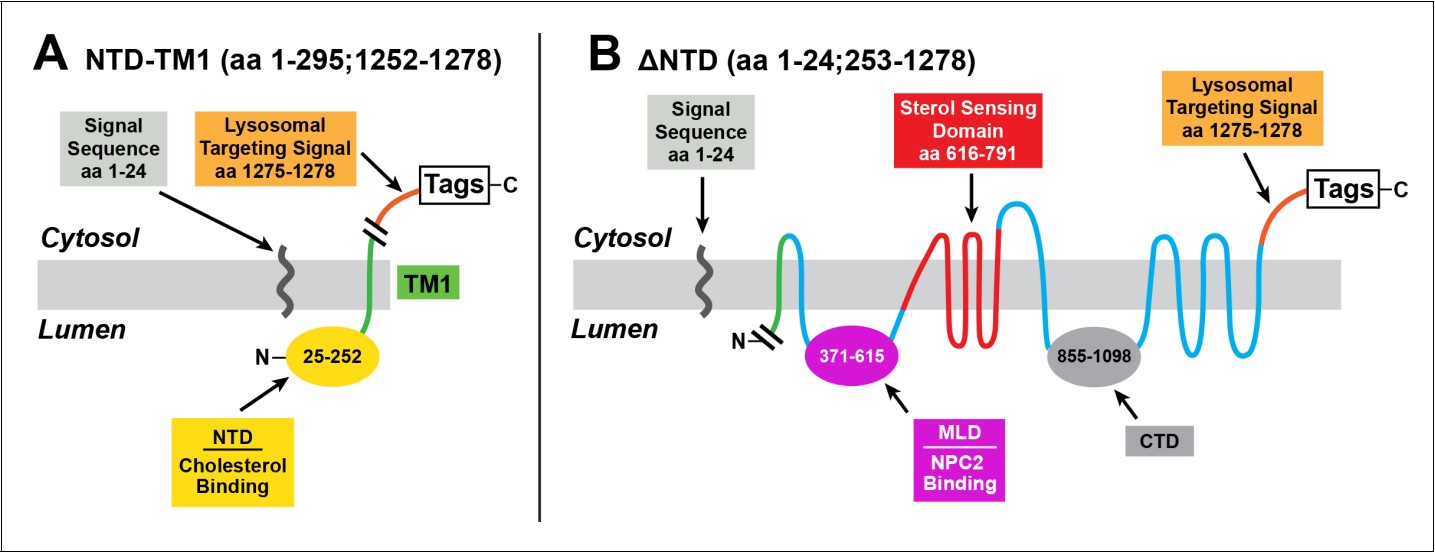

**Figure 2.** Mutant versions of human NPC1 used in cholesterol esterification assays. (A) pNTD-TM1 encodes the signal sequence of human NPC1, followed sequentially by the NTD, TM1, a lysosomal targeting signal, and epitope tags. (B) pΔNTD encodes NPC1 with a deletion of the NTD (amino acids 25–252). The cleaved signal sequence is shown.
DOI: https://doi.org/10.7554/eLife.38564.003

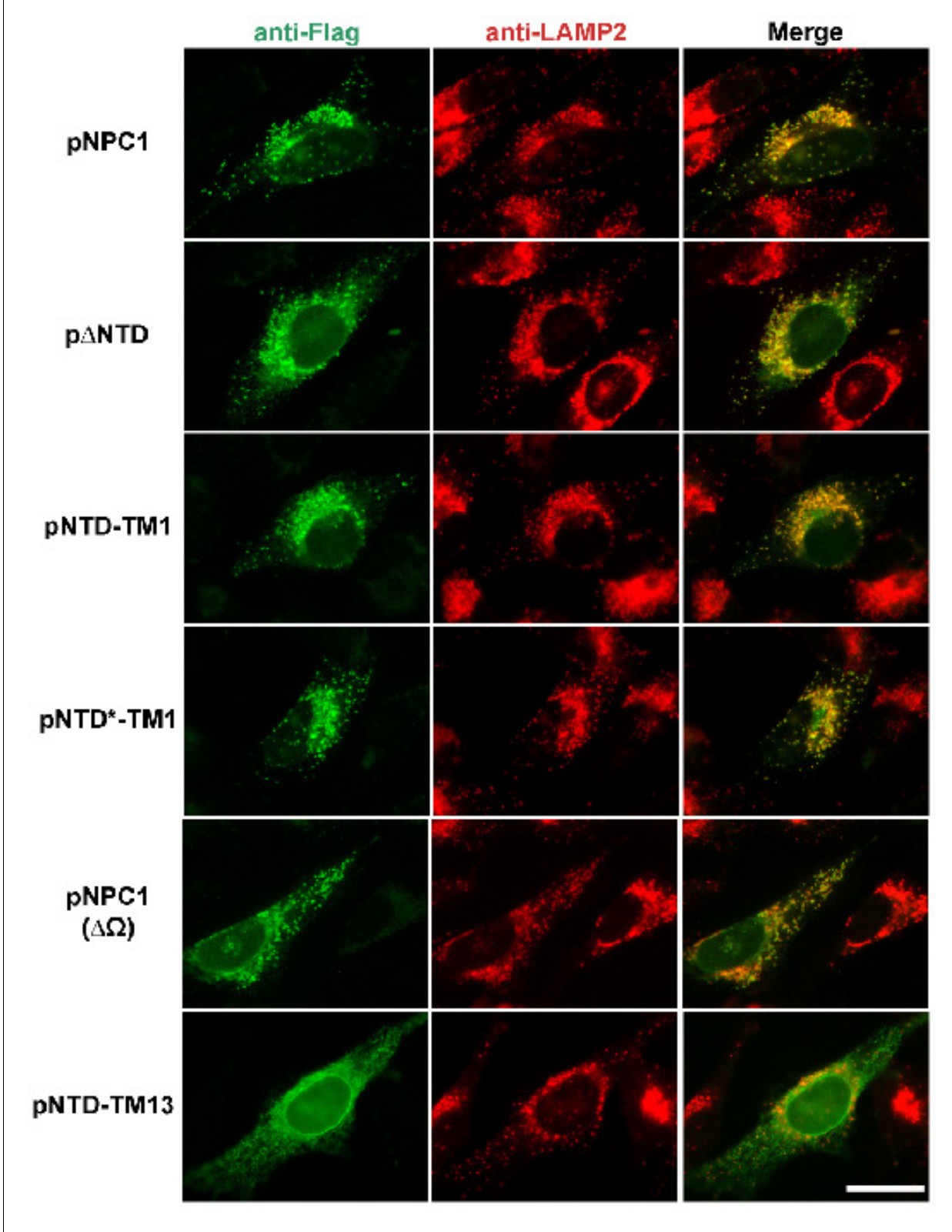

**Figure 3.** Localization of mutant NPC1 proteins to lysosomes. SV589 cells were transfected with the indicated plasmids encoding Flag-tagged fragments of NPC1 as described in Materials and methods. Cells were fixed and double stained with 0.8 μg/ml of rabbit monoclonal anti-Flag IgG (*green*) together with 1 μg/ml of mouse monoclonal anti-LAMP-2 IgG (*red*), and images were merged (*yellow*). LAMP-2 is a marker for lysosomes. Immunofluorescence microscopy was performed as described in Materials and methods. Scale bar, 20 μm.

*Figure 3 continued on next page*

*Figure 3 continued*

DOI: https://doi.org/10.7554/eLife.38564.004

versions of NPC1 that are described later in this paper. Two of these proteins localize to lysosomes. One of these, pNTD*-TM1, encodes NTD-TM1 with two amino acid substitutions that eliminate cholesterol binding (P202A/F203A), and the other, pNPC1(ΔΩ), encodes NPC1 with a deletion of the 9-amino acid loop (designated Ω) that mediates binding to the NTD (*Li et al., 2017*). The third version of NPC1, NTD-TM13, encodes the NTD followed by the 13th transmembrane helix of NPC1 instead of the first transmembrane helix; it shows a predominant ER localization.

To test the functional interaction between NTD-TM1 and ΔNTD, we used transfection to stably express ΔNTD in NPC1$^{-/-}$ cells, a clone of mutant CHO-K1 cells that lacks detectable NPC1 (*Wojtanik and Liscum, 2003*). To test for cholesterol transport from lysosomes, we incubated the cells either with lipoprotein-deficient serum or with FCS that contains LDL particles that deliver cholesterol to lysosomes through receptor-mediated endocytosis. When cholesterol is liberated from lysosomes, it travels to the endoplasmic reticulum (ER), where it is converted to cholesteryl esters by acyl-CoA:cholesterol acyltransferase (ACAT). Cholesterol esterification is quantified by adding [$^{14}$C] oleate to the medium and harvesting the cells for measurement of cholesteryl [$^{14}$C]oleate.

In the experiment of *Figure 4A*, we introduced full-length NPC1 into the NPC1$^{-/-}$ cells by transient transfection. When the cells were incubated with FCS, the LDL-derived cholesterol was transported out of lysosomes and esterified with [$^{14}$C]oleate (lane 2). No esterification was seen when we introduced varying amounts of pNTD-TM1, which encodes only the NTD of NPC1 (lanes 3–5). *Figure 4B* shows the result when we studied the permanent line of NPC1$^{-/-}$ cells that constitutively express ΔNTD. When we transfected increasing amounts of the NTD-TM1 plasmid, cholesterol transport was restored and cholesteryl [$^{14}$C]oleate was synthesized (lanes 8–10). This result indicates that the NTD can interact functionally with the remainder of NPC1 even when the NTD is not part of the same polypeptide chain. Expression of the endogenous ΔNTD and proteins produced by the transfected plasmids is shown in immunoblots in the bottom panel.

*Figure 5* shows a repeat of the complementation experiment of *Figure 4* using the line of NPC1$^{-/-}$ cells that stably express ΔNTD. Again, we found that these cells exhibit no cholesterol esterification when transfected with a control plasmid, pcDNA3.1. Esterification was restored when we transfected a plasmid encoding wild type NPC1 (*Figure 5A*). FCS-dependent esterification was also increased when the cells expressed NTD-TM1 together with ΔNTD (*Figure 5B*). Esterification was not restored when we expressed NTD-TM13 (*Figure 5C*), which shows a predominant ER rather than lysosomal localization (*see Figure 3*). Cholesterol transport was also not restored when we transfected the P202A/F203A mutant (pNTD*-TM1), which reaches the lysosome but does not function because it contains two point mutations that prevent cholesterol binding (*Figure 5D*). Expression of the endogenous ΔNTD and proteins produced by the transfected plasmids is shown in immunoblots in the bottom panel.

So far the data in this paper indicate that the NTD can transfer cholesterol to the remainder of NPC1 even when the NTD is attached to the membrane by only a single helix. We then sought to determine whether the NTD of one full-length NPC1 molecule can transfer its cholesterol to another full-length NPC1 molecule. To examine this question, we took advantage of the observation that the NTD appears to bind to a 9-amino acid loop that projects from the CTD as determined by a combined analysis of the structures determined by x-ray crystallography and cryo-electron microscopy (*Gong et al., 2016*; *Li et al., 2016b, 2017*). The loop is designated as the Ω loop. Deletion of the Ω loop abolished the ability of NPC1 to transport cholesterol out of the lysosome (*Li et al., 2017*). The protein lacking the Ω loop is designated NPC1(ΔΩ). Inasmuch as NPC1(ΔΩ) contains an intact NTD, we wondered whether that NTD could transfer its cholesterol to the ΔNTD protein that lacks an NTD but has an intact CTD (*see diagram in Figure 6A*).

We used the cholesterol esterification assay in NPC1$^{-/-}$ cells to assess the transfer of cholesterol from NPC1(ΔΩ) to ΔNTD (*Figure 6B*). When transfected with wild-type NPC1 and incubated with FCS, the NPC1$^{-/-}$ cells showed the expected cholesterol esterification (lanes 2–4). Esterification was not restored when the cells expressed NPC1(ΔΩ) (lanes 5–7). A different result was seen with the ΔNTD cells that stably express ΔNTD. In the absence of NPC1(ΔΩ), these cells showed no

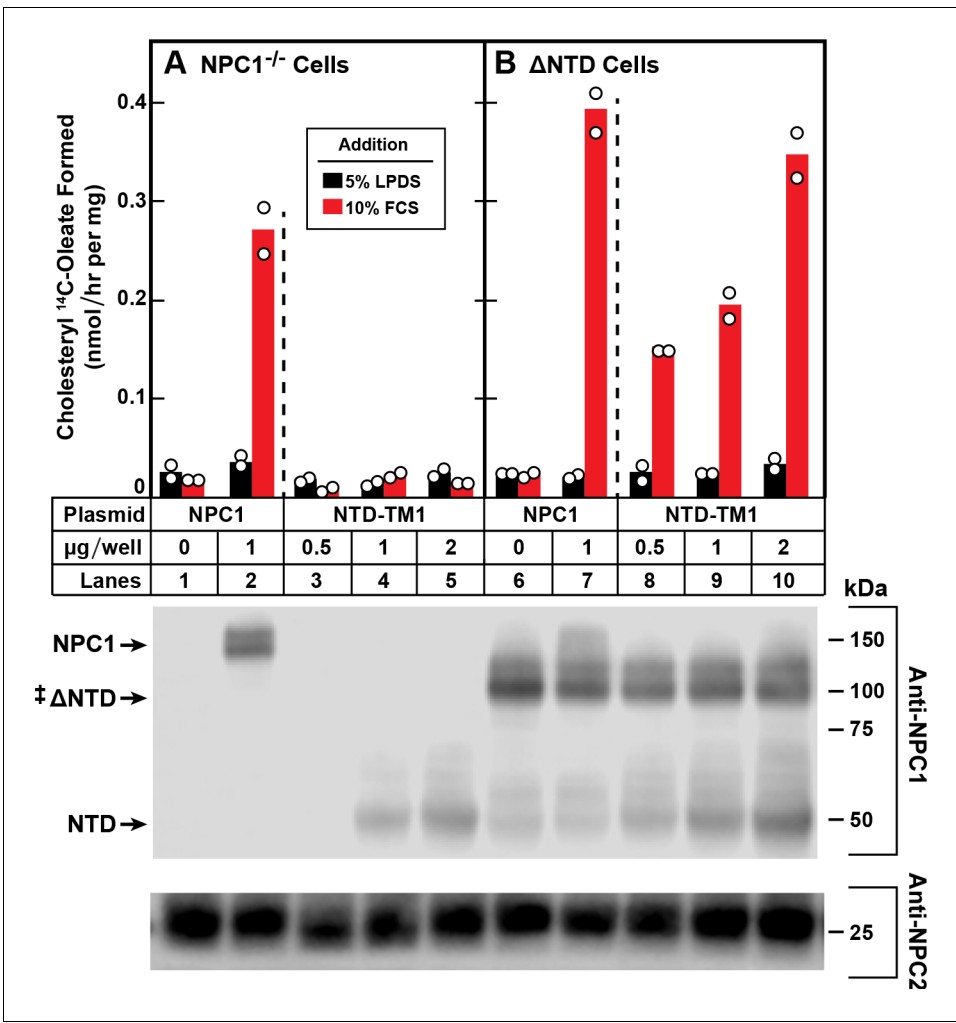

**Figure 4.** Trans-complementation between NTD and ΔNTD fragments of NPC1. On day 0, NPC1[-/-] cells (**A**) and NPC1[-/-] cells stably expressing ΔNTD (**B**) were set up in medium A with 5% FCS at $2.5 \times 10^5$ cells/60 mm dish. On day 1, monolayers were switched to fresh medium A (without antibiotics) with 5% LPDS and then transfected with the indicated plasmids encoding either full-length NPC1 or NTD-TM1. All dishes received FuGENE HD and a total of 2 μg DNA/dish adjusted with pcDNA3.1. After incubation for 24 hr, cells were washed once with PBS and switched to medium A with 5% LPDS containing 50 μM sodium compactin and 50 μM sodium mevalonate. On day 3, the cells received fresh medium B containing compactin and mevalonate in the presence of either 5% LPDS or 10% FCS containing lipoproteins. After incubation for 4 hr at 37°C, each cell monolayer was pulse-labeled for 2 hr with 0.1 mM sodium [14C]oleate (8568 dpm/nmol). The cells were then harvested for measurement of their content of cholesteryl [14C]oleate and [14C]triglycerides as described in Materials and methods. Each bar indicates the mean of duplicate incubations with individual values shown. The mean cellular content of [14C]triglycerides in the presence of FCS was not significantly different in NPC1[-/-] and ΔNTD cells (11.0 and 11.8 nmol/hr per mg protein, respectively). The bottom panel shows immunoblots of whole cell extracts (40 μg) using 0.36 μg/ml of rabbit monoclonal anti-NPC1 and 1.8 μg/ml of mouse monoclonal anti-NPC2. ‡ denotes the endogenous, stably transfected ΔNTD.

DOI: https://doi.org/10.7554/eLife.38564.005

cholesterol esterification (*Figure 6B*, *lane 8*). FCS-dependent esterification was enhanced when the cells co-expressed NPC1(ΔΩ) (lanes 12–14). Expression of the endogenous ΔNTD and the proteins produced by the transfected plasmids are shown in the bottom panel.

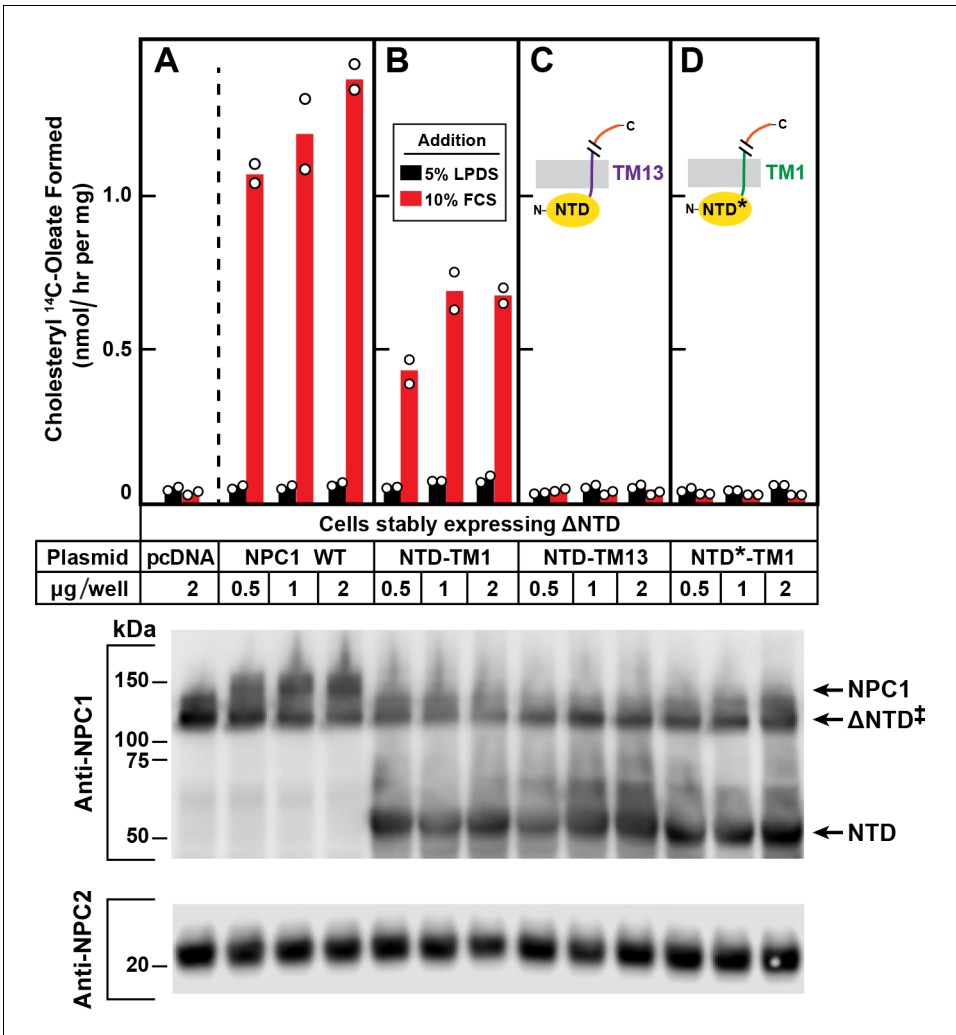

**Figure 5.** Restoration of cholesterol transport to ΔNTD sequences requires that the NTD of NPC1 localize to lysosomes and bind cholesterol. On day 0, ΔNTD cells were set up and transfected on day one as described in *Figure 4* with the indicated amount of one of the following plasmids: pcDNA3.1 (control) NPC1 (**A**), pNTD-TM1 (**B**), pNTD-TM13 (**C**), or pNTD*-TM1 (**D**). After incubation for 24 hr, cells were switched to medium A with 5% LPDS containing 50 μM sodium compactin and 50 μM sodium mevalonate. On day 3, the cells received fresh medium B containing compactin and mevalonate in the presence of either 5% LPDS or 10% FCS. After incubation for 4 hr at 37°C, each cell monolayer was pulse-labeled for 2 hr with 0.1 mM sodium [$^{14}$C]oleate (9019 dpm/nmol). The cells were then harvested for measurement of their content of cholesteryl [$^{14}$C]oleate and [$^{14}$C]triglycerides. Each bar indicates the mean of duplicate incubations with individual values shown. The mean cellular content of [$^{14}$C] triglycerides in the presence of FCS was not significantly different in cells transfected with pNPC1, pNTD-TM1, pNTD-T13, and pNTD*-TM1 (13.3, 12.7, 12.3, and 13.3 nmol per hr/mg protein, respectively). The bottom panel shows immunoblots of whole cell extracts (40 μg/lane) using 0.36 μg/ml of rabbit monoclonal anti-NPC1 and 1.8 μg/ml of mouse monoclonal anti-NPC2. ‡ denotes the endogenous, stably transfected ΔNTD.
DOI: https://doi.org/10.7554/eLife.38564.006

## Discussion

In order to exit the lysosome, LDL-derived cholesterol must pass three hurdles: (1) the insoluble sterol must be transported to the surface of the glycocalyx; (2) it must be transported across the 80 Å glycocalyx; and (3) it must be transported across the lipid bilayer that constitutes the lysosomal membrane. The first hurdle is overcome by NPC2, which binds cholesterol in the lumen and delivers it to the NTD of NPC1 that lies on the surface of the glycocalyx. The current study deals with the

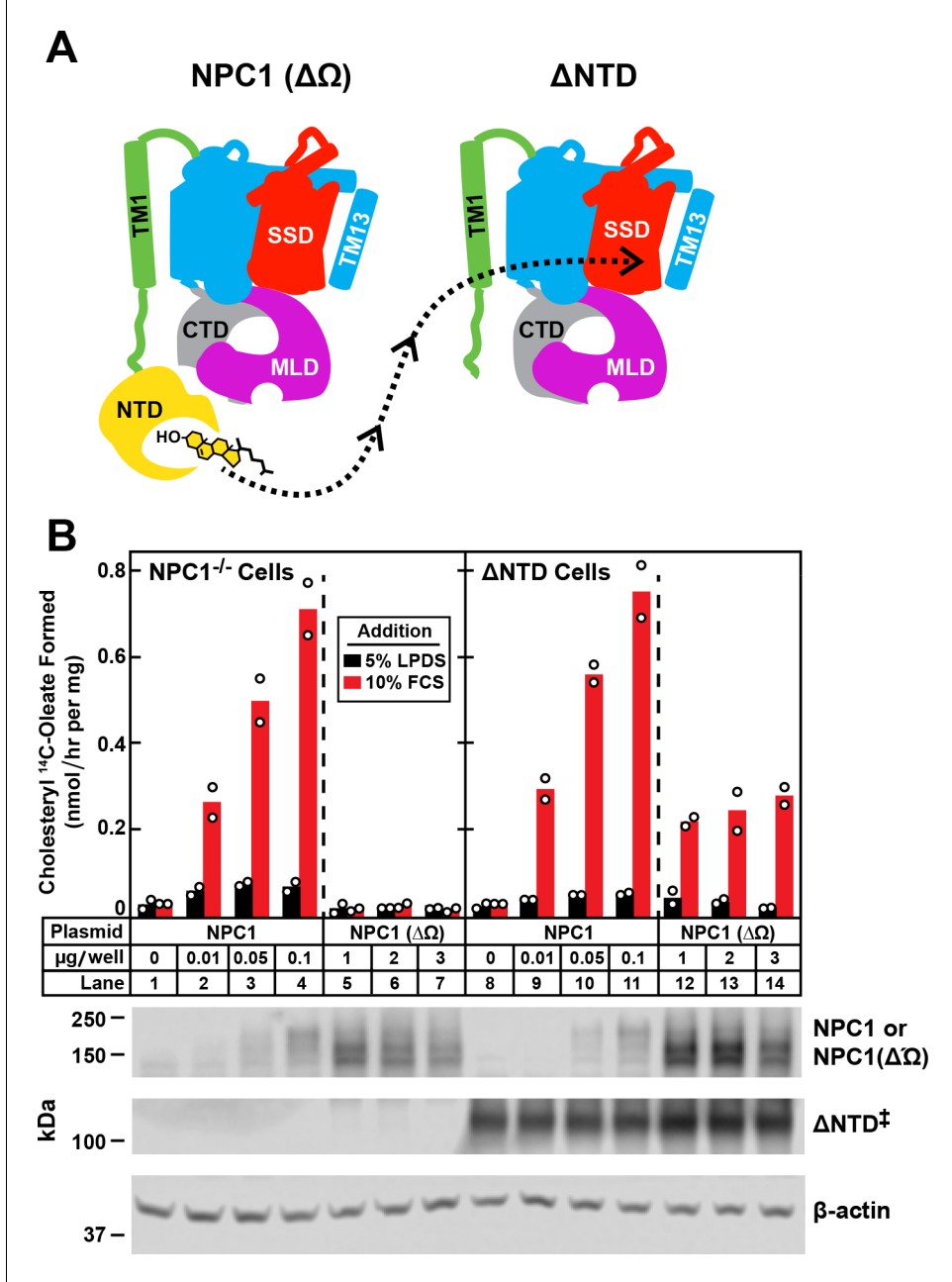

**Figure 6.** Transfer of cholesterol from NPC1(ΔΩ) to ΔNTD as determined by cholesterol esterification assay. (**A**) Model showing how the NTD of NPC1(ΔΩ) might transfer its cholesterol to the SSD of ΔNTD. (**B**) Cholesterol esterification assay. On day 0, NPC1[-/-] cells and ΔNTD cells were set up for experiments as described in *Figure 4* and transfected on day one with the indicated plasmids. All dishes contained a total of 3 μg of DNA adjusted with pcDNA3.1. After incubation for 24 hr, cells were washed once with PBS and switched to medium A with 5% LPDS containing 50 μM sodium compactin and 50 μM sodium mevalonate. On day 3, the cells received fresh medium B containing compactin and mevalonate in the presence of either 5% LPDS or 10% FCS. After incubation for 4 hr at 37°C, each cell monolayer was pulse-labeled for 2 hr with 0.1 mM sodium [14C]oleate (9452 dpm/nmol). The cells were then harvested for measurement of their content of cholesteryl [14C]oleate and [14C]triglycerides. Each bar indicates the mean of duplicate incubations with individual values shown. The mean cellular content of [14C] triglycerides in the presence of FCS was not significantly different in NPC1[-/-] and ΔNTD cells (11.8 and 14.8 nmol/ hr per mg protein, respectively). The bottom panel shows immunoblots of whole cell extracts (40 μg) using 3.6 μg/ ml rabbit polyclonal anti-NPC1(NTD) that detects both NPC1 ((lanes 2–4, 9–11) and NPC1 (ΔΩ) (lanes 5–7, 12–14),

*Figure 6 continued on next page*

*Figure 6 continued*

0.36 µg/ml rabbit monoclonal anti-NPC1 that detects ΔNTD* (lanes 8–14), and 0.2 µg/ml of mouse monoclonal anti-β-actin. ‡ denotes the endogenous, stably transfected ΔNTD.

DOI: https://doi.org/10.7554/eLife.38564.007

second hurdle – namely, the mechanism by which the NTD transports cholesterol across the glycocalyx.

Studies using cryo-EM demonstrated that the NTD is attached to a proline-rich stalk that is long enough to position the cholesterol-binding site just outside of the glycocalyx where it is accessible to NPC2 (*Gong et al., 2016*). In order for this cholesterol to reach the membrane, either the NTD must move by 80 Å to deliver it there or the cholesterol must be released to an intermediate carrier or channel that is formed by the MLD and CTD. No such channel is visible in the structures obtained by cryo-EM (*Gong et al., 2016*) or X-ray crystallography (*Li et al., 2016b, 2017*).

In the current paper, we demonstrate that the NTD can transfer its cholesterol to the membrane even when the NTD is not covalently attached to the remainder of NPC1. This finding suggests that any change in the location of the NTD is mediated by its interactions with the MLD and CTD. We also show that the NTD of one full-length NPC1 can transfer its cholesterol to another NPC1 that lacks its own NTD. This finding suggests that full-length NPC1 molecules interact with each other, raising the possibility that NPC1 molecules form a complex with each other and that cholesterol transfer results from interactions between neighboring molecules.

NPC1 is related by amino acid sequence and structural conformation to a family of bacterial proteins called resistance-nodulation-cell division (RND) proteins (*Tseng et al., 1999*). These proteins function as proton-coupled antiporters, expelling a variety of structurally unrelated hydrophobic toxins from gram negative bacteria (*Seeger et al., 2006*; *Yamaguchi et al., 2015*). Like NPC1, RND proteins must transport their cargo across a lipid membrane and across an aqueous space (the periplasm). When purified in detergents, RND proteins crystallize as homotrimers. One of the monomers binds the ligand and undergoes a series of conformational changes that shifts the ligand through a protein-enclosed tunnel until it is discharged on the other side.

In contrast to RND proteins, NPC1 is a monomer when purified in detergents and subjected to cryo-EM or X-ray crystallography. Nevertheless, the current evidence of interaction between neighboring NPC1 molecules raises the possibility that the protein may form functional multimers in the membrane and that cholesterol may be transported through the glycocalyx by means of a functional interaction between neighboring NPC1 molecules.

## Materials and methods

### Key resources table

| Reagent type (species) or resource | Designation | Source or reference | Identifiers | Additional information |
|---|---|---|---|---|
| Chemical compound | Sodium dodecyl sulfate (SDS) | Sigma-Aldrich | 71736 | |
| Chemical compound | Benzonase nuclease | Sigma-Aldrich | E1014 | |
| Chemical compound | Bovine serum albumin | Sigma-Aldrich | A7284 | |
| Chemical compound | [1-$^{14}$C]Oleic acid (50 mCi/mmol) | PerkinElmer, Waltham, MA | NEC317050UC | |
| Chemical compound | SuperSignal West Pico Chemiluminescent Substrate | Thermo Fisher Scientific | 34580 | |
| Chemical compound | Zeocin | Life Technologies, Grand Island, NY | R25005 | |
| Chemical compound | FuGENE HD | Promega Corporation, Madison, WI | E2311 | |
| Chemical compound | Formaldehyde | Sigma-Aldrich | F8775 | |

*Continued on next page*

*Continued*

| Reagent type (species) or resource | Designation | Source or reference | Identifiers | Additional information |
|---|---|---|---|---|
| Chemical compound | Penicillin-Streptomycin Solution | Corning | 30–002 Cl | |
| Chemical compound | Methanol | Fisher Scientific, Hampton, NH | A412 | |
| Chemical compound | Hexane | Fisher Scientific | H292 | |
| Chemical compound | Isopropanol | Fisher Scientific | A416 | |
| Chemical compound | Heptane | Fisher Scientific | H350 | |
| Chemical compound | Ethyl ether | Fisher Scientific | E138 | |
| Chemical compound | Acetic acid | Fisher Scientific | A38C | |
| Chemical compound | Sodium compactin | *Brown et al. (1978)* | NA | |
| Chemical compound | Sodium mevalonate | *Brown et al. (1978)* | NA | |
| Other | L-Glutamine-free DMEM | Sigma-Aldrich | D5546 | culture medium |
| Other | DMEM-low glucose (1000 mg/l) | Sigma-Aldrich | D6046 | culture medium |
| Other | Ham's F-12 medium and Dulbecco's modified Eagle's medium containing 2.5 mM L-glutamine (DMEM) | Corning, Manassas, VA | 10–090-CV | culture medium |
| Other | Newborn calf lipoprotein-deficient serum (LPDS, d < 1.215 g/mL) | *Goldstein et al. (1983)* | NA | culture serum |
| Commercial assay or kit | Bolt 4–12% Bis-Tris Plus gradient gels | Thermo Fisher Scientific, Waltham, MA | NW04125BOX | |
| Commercial assay or kit | QuikChange II XL Site-Directed Mutagenesis Kit | Agilent Technologies, Santa Clara, CA | 200522 | |
| Antibody | Rabbit monoclonal IgG against Flag | Sigma-Aldrich, St. Louis, MO | F7425, RRID: AB_439687 | |
| Antibody | Mouse monoclonal IgG against LAMP-2 | BD Biosciences, Franklin Lakes, NJ | 555803, RRID: AB_396137 | |
| Antibody | Rabbit monoclonal IgG against amino acids 1261–1278 of human NPC1 | Abcam, Cambridge, UK | ab134113 | |
| Antibody | Rabbit polyclonal IgG against NPC1(NTD)-His8-FLAG | *Infante et al. (2008)* | Clone 491B | |
| Antibody | Goat anti-rabbit IgG conjugated to AlexaFluor 488 | Invitrogen, Carlsbad, CA | A-11008, RRID: AB_143165 | |
| Antibody | Goat anti-mouse IgG conjugated to AlexaFluor 594 | Invitrogen | A-11005, RRID: AB_141372 | |
| Antibody | Mouse monoclonal HRP-conjugated IgG against β-actin | Cell Signaling Technology, Danvers, MA | 12262, RRID: AB_2566811 | |
| Antibody | Horse anti-mouse IgG conjugated to HRP | Cell Signaling Technology | 7076, RRID: AB_330924 | |
| Antibody | Goat anti-rabbit IgG conjugated to HRP | Cell Signaling Technology | 7074, RRID: AB_2099233 | |

*Continued on next page*

*Continued*

| Reagent type (species) or resource | Designation | Source or reference | Identifiers | Additional information |
| --- | --- | --- | --- | --- |
| Antibody | Mouse monoclonal IgG against human NPC2 | *Wang et al. (2010)* | Clone 13G4 | |

## Culture media

Medium A is a 1:1 mixture of Ham's F-12 medium and Dulbecco's modified Eagle's medium (DMEM) containing 2.5 mM L-glutamine. Medium B is L-glutamine-free DMEM. Medium C is DMEM-low glucose (1000 mg/l). All media contained 100 units/ml penicillin and 100 µg/ml streptomycin sulfate unless otherwise noted.

## Cell culture

NPC1$^{-/-}$ cells (previously referred to as 10–3 cells) are a stable line of mutant CHO-K1 cells that lack detectable NPC1 (*Wojtanik and Liscum, 2003*). These NPC1$^{-/-}$ cells and CHO-K1 cells were grown in medium A with 5% FCS. ΔNTD cells are a stable cell line of mutant NPC1$^{-/-}$ cells that stably express human NPC1 lacking the N-terminal domain (described below). These cells were grown in medium A with 5% FCS and 500 µg/ml Zeocin. CHO-7 cells, a clone of CHO-K1 cells selected for growth in LPDS (*Metherall et al., 1989*), were grown in medium A with 5% LPDS. SV589 cells are a line of SV40-immortalized human fibroblasts (*Yamamoto et al., 1984*). These cells were grown in medium C with 5% FCS.

Stock cultures of all cell lines were maintained in monolayer culture at 37°C in an 8.8% CO$_2$ incubator except for the SV589 cells, which were maintained at 5% CO$_2$. All cell lines were routinely monitored for mycoplasma contamination.

## Plasmid constructions

pNPC1 encodes wild-type human NPC1 (amino acids 1–1278) followed sequentially by three tandem copies of the Flag epitope tag (DYKDDDDK), one copy of the TEV cleavage site (ENLYFQ), and two copies of the StrepTactin epitope tag (WSHPQFEK). Expression is achieved with the *cytomegalovirus* (CMV) promoter. This plasmid was constructed by ligating the component DNA sequences into the 5′-XbaI and 3′-HindIII sites of pcDNA3.1/Zeo(-) (*Lu et al., 2015*). The original plasmid used to generate pNPC1 was constructed from pCMV-NPC1 (Origene Technologies, Rockville, MD). Deletions and point mutations were introduced into the coding region of pNPC1 by site-directed mutagenesis using the QuikChange II XL Site-Directed Mutagenesis Kit (Agilent Technologies, Santa Clara, CA).

pΔNTD encodes NPC1 with a deletion of amino acids 25–252. pNTD-TM1 encodes NPC1 with a deletion of amino acids 296–1251 followed by a lysosomal targeting sequence (*Watari et al., 1999*) and two epitope tags (Flag and StepTactin) as described above for pNPC1. pNTD-TM13 encodes NPC1 with a deletion of amino acids 260–1217 followed by the same sequences added to pNTD-TM1. pNTD*-TM1 encodes the NTD of NPC1 with a double point mutation (P202A/F203A) followed by the same sequences added to pNTD-TM1. pNPC1(ΔΩ) encodes NPC1 in which a amino acids 909–917 are deleted and replaced with a single alanine residue.

The coding region of each plasmid was sequenced to ensure integrity of the construct.

## Generation of ΔNTD cells that lack the N-terminal domain of NPC1

NPC1$^{-/-}$ CHO-K1 cells were set up on day 0 at a density of $4 \times 10^5$ cells per 100 mm dish in 10 ml of medium A with 5% LPDS. On day 2, cells were transfected with 1 µg/dish of pΔNTD using FuGENE HD transfection reagent according to the manufacturer's instructions. At 24 hr after transfection, 700 µg/ml Zeocin was added for selection. Fresh medium was added every 2–3 days until colonies formed at ~15 days. Individual colonies were isolated with cloning cylinders, and expression of ΔNTD was assessed by immunoblot analysis with rabbit monoclonal anti-NPC1. Cells from single colonies were cloned by limiting dilution, maintained in medium A with 5% LPDS containing 500 µg/ml Zeocin and are hereafter referred to as ΔNTD cells.

## Immunoblot analysis

Whole cell extracts were subjected to electrophoresis in phosphate-buffered saline (PBS) containing 0.25% SDS and a 1:1000 dilution of Benzonase Nuclease. Samples were applied to Bolt 4–12% gradient gels. After electrophoresis, the proteins were transferred to nitrocellulose filters, which were then incubated with the indicated primary antibody (see figure legends). Bound antibodies were visualized by chemiluminescence (SuperSignal West Pico Chemiluminescent Substrate, Thermo Scientific, Waltham, MA) after a 1 hr incubation with either 31 ng/ml of horse anti-mouse IgG or 13 ng/ml of goat anti-rabbit IgG conjugated to horseradish peroxidase. The immunoblot using the HRP-conjugated β-actin antibody was visualized without a secondary antibody. The images were scanned using an Odyssey FC Imager (Dual-Mode Imaging System; 2 min integration time) and analyzed using Image Studio ver. 5.0 (LI-COR Biosciences, Lincoln, NE).

## Cholesterol esterification assay

The rate of incorporation of [$^{14}$C]oleate into cholesteryl [$^{14}$C]oleate and [$^{14}$C]triglycerides by monolayers of NPC1$^{-/-}$ cells and ΔNTD cells was measured as described previously (*Metherall et al., 1989*). The details of cell plating, incubation conditions, transfections, and pulse labeling with [$^{14}$C] oleate are described in the figure legends. After a 4 hr incubation with [$^{14}$C]oleate, the cells were washed, and the lipids were extracted in hexane:isopropanol (3:2, vol:vol), separated on a silica gel G thin-layer chromatogram (developed in heptane:ethylether:acetic acid, 90:30:1), and quantified by scintillation counting with the use of an internal standard for recovery (*Goldstein et al., 1983*). The amounts of cholesteryl [$^{14}$C]oleate and [$^{14}$C]triglycerides formed are expressed as nanomoles formed per hour per milligram cell protein.

## Co-localization by immunofluorescence microscopy

SV589 cells were set up on glass coverslips at $1.5 \times 10^5$ cells per 6-well plate in 2 ml medium C with 5% FCS. At 24 hr after plating, cells were transfected with 1 μg of the indicated plasmid using FuGENE HD as the transfection agent. At 24 hr after transfection, cells were fixed for 15 min in 3.7% formaldehyde in PBS at room temperature and permeabilized for 10 min in methanol at −20°C. After blocking by incubation with 1 mg/ml bovine serum albumin in PBS, cells were double-labeled with 1 μg/ml mouse monoclonal anti-LAMP-2 and 0.8 μg/ml of rabbit monoclonal anti-Flag followed by 6.7 μg/ml goat anti-rabbit IgG conjugated with AlexaFluor 488 and 6.7 μg/ml goat anti-mouse IgG conjugated with AlexaFluor 594. The coverslips were than mounted in Mowiol (EMD, Darmstadt, Germany) solution (*Wei and Seemann, 2009*) and fluorescence images were acquired using a Plan-Neofluar 40x/1.3 DIC objective (Zeiss, Oberkochen, Germany), an Axiovert 200M microscope (Zeiss), an Orca 285 camera (Hamamatsu, Hamamatsu City, Japan), and Openlab 4.0.2 software (Improvision, Coventry, UK).

## Reproducibility

All experiments were repeated three or four times on different days. Similar results were obtained.

# Acknowledgments

We thank our colleagues Ting Han, Xiaochun Li, and Daniel Rosenbaum for helpful suggestions; Ian Ford and Jessica Proulx for excellent technical assistance; and Lisa Beatty and Shomanike Head for invaluable help with tissue culture.

# Additional information

### Funding

| Funder | Grant reference number | Author |
|---|---|---|
| National Institutes of Health | HL20948 | Michael S Brown<br>Joseph L Goldstein |
| Welch Foundation | I-1910 | Joachim Seemann |
| National Institutes of Health | T32 GM008014 | Michael Nguyen Trinh |

| National Institutes of Health | GM096070 | Joachim Seemann |
|---|---|---|

The funders had no role in study design, data collection and interpretation, or the decision to submit the work for publication.

## Author contributions
Michael Nguyen Trinh, Feiran Lu, Designed research, Performed research, Wrote the paper, Analyzed data, Approved the final version of the paper; Michael S Brown, Joseph L Goldstein, Designed research, Wrote the paper, Analyzed data, Approved the final version of the paper; Joachim Seemann, Performed research, Analyzed data, Approved the final version of the paper

## Author ORCIDs
Michael Nguyen Trinh (iD) https://orcid.org/0000-0003-0600-1349
Joseph L Goldstein (iD) https://orcid.org/0000-0002-1894-9463
Feiran Lu (iD) https://orcid.org/0000-0002-1757-2895

## Decision letter and Author response
Decision letter https://doi.org/10.7554/eLife.38564.012
Author response https://doi.org/10.7554/eLife.38564.013

# Additional files

## Supplementary files
• Transparent reporting form
DOI: https://doi.org/10.7554/eLife.38564.008

## Data availability
All data generated or analyzed during this study are included in the manuscript and supporting files. Source data files have been provided for Figures 1 and 6.

The following previously published datasets were used:

| Author(s) | Year | Dataset title | Dataset URL | Database, license, and accessibility information |
|---|---|---|---|---|
| Gong X, Qian HW, Zhou XH, Wu JP, Zhou Q, Yan N | 2016 | Cryo-EM Structure of the Full-Length human NPC1 at 4.4 Angstrom | http://www.rcsb.org/structure/3JD8 | Publicly available at RCSB Protein Data Bank (accession no. 3JD8) |

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
