## [Decision Letter]

Thank you for submitting your article "Lysosomal cholesterol export reconstituted from fragments of Niemann-Pick C1" for consideration by *eLife*. Your article has been reviewed by three peer reviewers, and the evaluation has been overseen by a Reviewing Editor and Randy Schekman as the Senior Editor. The following individual involved in review of your submission has agreed to reveal his identity: Hongyuan Yang (Reviewer #3).

The reviewers have discussed the reviews with one another and the Reviewing Editor has drafted this decision to help you prepare a revised submission.

Summary:

This study investigates the mechanism of transport of LDL-derived cholesterol out of lysosomes. Cholesteryl esters in LDL are hydrolyzed in lysosomes and the free cholesterol is then transferred by NPC2 and NPC1 to the lysosome membrane, where it is trafficking out of lysosomes. Current models suggest that cholesterol is first delivered to the N-terminal domain and then handed off to a distant, second sterol binding site adjacent to the membrane. Here the authors show that a membrane anchored form of the N-terminal domain (NTD) can transfer cholesterol to the rest of another molecule lacking the NTD or even from one full length molecule with a nonfunctional sterol binding domain to another molecule with a non-functional NTD. Importantly, this study suggests that the NTD cannot transfer cholesterol to the membrane alone. In addition, a split nanoluciferase assay to show that two NPC1 molecules or the NTD and the rest of NPC1 can interact. This study advances our understanding of how the NTD functions, though mechanistic details remain to be determined.

Essential revisions:

1) It is important to acknowledge and discuss the limitations of using split nanoluciferase to assess whether two proteins interact. Specifically, the possibility that the reporter may drive interactions should be discussed.

2) The control TM domain for the split nanoluciferase assays is not in lysosomes and thus is not an appropriate control. It is necessary to use a control that efficiently targets lysosomes and is not expected to interact with NPC1.

3) It is important to verify that the split nanoluciferase constructs traffic to lysosomes.

4) The relative expression levels of the complementing constructs should be provided.

5) The results in Figure 9B should be shown without a broken axis. It is also important to note that the results in Figure 9B do not necessarily indicate that the NTD reaches through the glycocalyx.

6) Determine whether the interaction of ΔNTD and ΔΩ is cholesterol dependent, similar to what was shown for NTD-TM1 in Figure 8.

---

## [Author Response]

Summary:This study investigates the mechanism of transport of LDL-derived cholesterol out of lysosomes. Cholesteryl esters in LDL are hydrolyzed in lysosomes and the free cholesterol is then transferred by NPC2 and NPC1 to the lysosome membrane, where it is trafficking out of lysosomes. Current models suggest that cholesterol is first delivered to the N-terminal domain and then handed off to a distant, second sterol binding site adjacent to the membrane. Here the authors show that a membrane anchored form of the N-terminal domain (NTD) can transfer cholesterol to the rest of another molecule lacking the NTD or even from one full length molecule with a nonfunctional sterol binding domain to another molecule with a non-functional NTD. Importantly, this study suggests that the NTD cannot transfer cholesterol to the membrane alone. In addition, a split nanoluciferase assay to show that two NPC1 molecules or the NTD and the rest of NPC1 can interact. This study advances our understanding of how the NTD functions, though mechanistic details remain to be determined.

We thank you and the reviewers for your careful review of our manuscript entitled "Lysosomal cholesterol export reconstituted from fragments of Niemann-Pick Cl." The main point of the paper derives from functional studies showing that cholesterol transport from lysosomes can be reconstituted from two fragments of NPCI, indicating that the cholesterol-binding N-terminal domain can transfer its cholesterol to the membrane-embedded portion of NPCI even when the Nterminal domain is not linked covalently to the membrane portion. In addition, we show that cholesterol can be transferred from the NTD of one NPCI to the transport domain of a second NPC 1, indicating that NPC I may function as a multimer, a hypothesis that has not been suggested previously. As an additional element, the manuscript also employed a split nanoluciferase reconstitution assay to demonstrate that the N-terminal domain binds to the remaining portion of NPC 1.

The reviewers did not express significant concerns about the functional reconstitution as measured by cholesterol esterification. However, they raised several questions regarding the split nanoluciferase interaction. We recognize that additional controls would be necessary to verify the specificity of the split nanoluciferase assay. Performing these controls will require the development of several lines of stably transfected cells that will take many months to complete. Even then, the results may not rule out the possibility that the interaction is caused by the binding of the two luciferase fragments instead of the binding of NPC1 fragments.

We believe that the functional reconstitution assay stands on its own and provides important new information for this field. Accordingly, we have eliminated all figures and all text references that deal with the split nanoluciferase assay. Inasmuch as there were no expressed reservations regarding the functional reconstitution assays, we hope that this manuscript will now be acceptable for publication in *eLife*.

Essential revisions:1) It is important to acknowledge and discuss the limitations of using split nanoluciferase to assess whether two proteins interact. Specifically, the possibility that the reporter may drive interactions should be discussed.2) The control TM domain for the split nanoluciferase assays is not in lysosomes and thus is not an appropriate control. It is necessary to use a control that efficiently targets lysosomes and is not expected to interact with NPC1.3) It is important to verify that the split nanoluciferase constructs traffic to lysosomes.

These comments (1 – 3) relate to the nanoluciferase experiments that have now been removed from the revised paper. We have removed Figure 6, Figure 7, Figure 8, and Figure 9C, D.

4) The relative expression levels of the complementing constructs should be provided.

The relative expression levels of the complementing constructs in Figure 4 have now been assessed by immunoblotting and added to Figure 4 (bottom panel).

5) The results in Figure 9B should be shown without a broken axis. It is also important to note that the results in Figure 9B do not necessarily indicate that the NTD reaches through the glycocalyx.

The original Figure 9B is now Figure 6B, which has been repeated as a new experiment to include immunoblots for protein expression. In the new figure, there is no broken axis.

Since we do not mention the glycocalyx in our discussion of the original Figure 9B (now Figure 6B), we do not understand the relevance of this comment. The evidence that the NTD reaches through the glycocalyx is based on the structure of NPCI (see Figure 1B).

6) Determine whether the interaction of ΔNTD and ΔΩ is cholesterol dependent, similar to what was shown for NTD-TM1 in Figure 8.

This comment relates to a nanoluciferase experiment that has been removed from the revised paper.